# Are Language Models Robust Coreference Resolvers?

**Nghia T. Le & Alan Ritter**
Georgia Institute of Technology
Atlanta, GA 30332, USA
{nle18,alan.ritter}@cc.gatech.edu

## Abstract

Recent work on extending coreference resolution across domains and languages relies on annotated data in both the target domain and language (Xia & Van Durme, 2021). At the same time, pre-trained large language models (LMs) exhibit strong zero- and few-shot learning abilities across a wide range of NLP tasks. However, prior work mostly studied this ability using artificial sentence-level datasets such as the Winograd Schema Challenge. In this paper, we assess the feasibility of prompt-based coreference resolution by evaluating instruction-tuned language models on difficult, linguistically-complex coreference benchmarks (e.g., CoNLL-2012). We show that prompting for coreference can outperform current unsupervised coreference systems, although this approach appears to be reliant on high-quality mention detectors. Further investigations reveal that instruction-tuned LMs generalize surprisingly well across domains, languages, and time periods; yet continued fine-tuning of neural models should still be preferred if small amounts of annotated examples are available. [1]

## 1 Introduction

Entity coreference resolution aims to find all spans within an input text that refer to the same entity. As an important information extraction sub-task, coreference resolution has received considerable attention from the NLP community over the years, with recent progress driven mostly by neural coreference models (Lee et al., 2017; Wu et al., 2020; Joshi et al., 2020). There has also been an increasing interest in the generalization of coreference systems to domains and languages beyond the popular CoNLL-2012 benchmark (Xia & Van Durme, 2021; Bohnet et al., 2022). Most work on extending coreference resolution to new domains and languages relies on target language annotated data in the targeted domain, however the amount of labeled data needed to cover every possible domain in all languages is prohibitively expensive. Meanwhile, unsupervised (Haghighi & Klein, 2010) and few-shot (Le et al., 2022) coreference resolution has received less attention, despite the fact that learning with less labels is desirable when adapting to new languages or domains.

Concurrently, there has been a great deal of progress on zero- and few-shot learning using pre-trained language models (LMs) (Ouyang et al., 2022; Touvron et al., 2023). Attempts have been made at evaluating pre-trained LMs' coreference abilities under zero- and few-shot settings: Brown et al. (2020) demonstrated that prompting GPT-3 can resolve coreference on the Winograd Schema Challenges (WSC), Yang et al. (2022) showed that coreference resolution was a challenging task for GPT-2 when prompted with multiple-choice templates, and Agrawal et al. (2022) successfully reframed clinical pronoun resolution as span generation. While these studies reveal some evidence of the coreference abilities in large LMs, they either use methods that fail to beat reasonable baselines, or evaluate on sentence-level, non-standard coreference datasets that are designed to benchmark the capabilities of LLMs, rather than providing an accurate evaluation of models' coreference capabilities in a realistic setting. In contrast, the traditional dataset for coreference resolution, CoNLL-2012/OntoNotes, contains real-world document-level examples with complex linguistic

---

[1]Our code is available at https://github.com/nle18/coref-llms

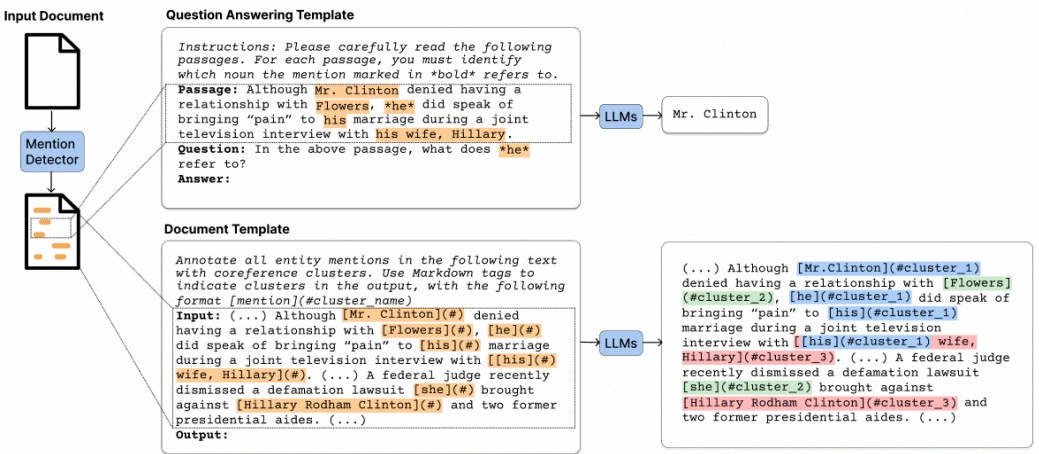

Figure 1: An example of coreference resolution with LMs prompting. Here we show two prompt templates experimented in this work: Question-Answer and Document templates. In the QA template, the language model generates the answer when given a passage and an open-ended *wh*-question (Ouyang et al., 2022). In contrast, the document template marks the candidate mentions and asks the LM to annotate the cluster IDs for each mention directly within the text (represented by different colors). Both templates require a mention detector to generate candidate mentions.

annotations (Pradhan et al., 2012). Evaluating LMs using OntoNotes' realistic inputs is arguably more suitable for the evaluation of LMs' coreference capabilities.

In this paper, we aim to bridge the gap between the coreference and language modeling literature by investigating to what extent instruction-tuned language models can perform coreference resolution via prompting. We show that prompting LMs is a feasible strategy for coreference resolution, outperforming previous unsupervised systems. Nonetheless, it still trails behind state-of-the-art supervised models and relies heavily on a robust mention detector. Finally, we explore the generalization ability of this approach by extending our analysis to a diverse range of domains, languages, and time periods. Our results indicate that fine-tuning should still be the preferred option if a large out-of-domain corpus and a few annotated in-domain documents are available. However, large instruction-tuned LMs can generalize surprisingly well across domains and languages, making them a robust option if no target language or in-domain data is available for fine-tuning.

**Contributions** Our main contributions are: (a) a simple yet effective LM prompting method for coreference resolution that outperforms SOTA unsupervised baselines, (b) a systematic comparison analysis of the strengths and weaknesses of prompting LMs for coreference against strong baselines, and (c) we empirically demonstrate the robust generalization ability of LMs for coreference across different domains, languages, and time periods.

## 2   Prompt-based Coreference Resolution

Previous work in zero- and few-shot coreference resolution assumes access to candidate mentions to resolve (Ouyang et al., 2022; Agrawal et al., 2022). We adopt this formulation: given a document, and a set of candidate mentions (gold or predicted), we prompt an autoregressive language model, and extract the predicted coreference links (Figure 1).

Prior work applying language models to resolve co-referring entity mentions has mainly experimented with Question-Answer (QA) prompts for pronoun resolution (Ouyang et al., 2022; Agrawal et al., 2022) and demonstrated its effectiveness when comparing with other templates such as multiple-choice (Arora et al., 2022). However, in a preliminary study

(§A.1), we found that prompting GPT-4 with a QA template struggled to compete with Stanford's deterministic coreference system (Lee et al., 2013), achieving 67 F$_1$ when comparing to 72 F$_1$ from Lee et al. (2013). We also experimented with an alternative document-level template that can elicit more coreference links than the traditional QA template, achieving 86 F$_1$ (Table A.1). In this template, the mentions of the input text are first marked with special tokens indicating a span to annotate (e.g., *Mr. Clinton → [Mr. Clinton](#)*). The LM is then given instructions to annotate this marked span with the cluster ID, (e.g., *[Mr. Clinton](#) → [Mr. Clinton](#cluster_1)*). Given strong results over the QA template, we used this document template for all subsequent experiments.

## 3   CoNLL-2012 Experiments

We investigate the coreference abilities of large LMs on the CoNLL-2012 benchmark (Pradhan et al., 2012) and compare these models against existing supervised and unsupervised baselines. We found that GPT model family (OpenAI, 2023) yield competitive results with previous unsupervised and rule-based models, while significantly outperforming them when gold mentions are provided.

### 3.1   Experimental Details

**Dataset and Evaluation Metrics**   The English OntoNotes 5.0 dataset (Weischedel et al., 2011; Pradhan et al., 2012) is traditionally used to evaluate coreference systems. This dataset spans seven distinct genres such as news, telephone conversations, and religious text. We follow the standard train-dev-test splits from previous work and report CoNLL F$_1$, an average of three coreference-based metrics MUC, B$^3$, and CEAF$_{\phi_4}$.

**Baselines**   We mainly consider Stanford's deterministic resolver, which is referred to as dcoref (Lee et al., 2013). This coreference resolver consists of multiple sieves, with each sieve being a set of handcrafted rules that filters out mentions and ordered from highest to lowest precision to minimize cascading errors. [2] For supervised systems, we compare to coref-mt5 (Bohnet et al., 2022) and coref-T0 (Zhang et al., 2023), two text-to-text approaches based on seq2seq models. For unsupervised baselines, we include results from weak-SpanBERT (Stolfo et al., 2022), a system that trained a SpanBERT-based architecture on dcoref coreference predictions.

**Open-sourced Models**   We use Llama 2 model family (Touvron et al., 2023) as the primary open-sourced language models, namely two instruction-tuned versions of base Llama-2, Llama-2-Chat and CodeLlama (Rozière et al., 2023). To avoid hallucinations, we constrain the generations as follows: for each given mention, we ask the model to generate the cluster ID. We then update the input sequence by appending the generated ID with the text segment between the current and the next mention. The process is repeated until all the mentions in the document are annotated.

**Proprietary Models**   We also report performance on the most recent OpenAI language models: the instruction-tuned 175B InstructGPT (text-davinci-003) (Ouyang et al., 2022), ChatGPT (gpt-35-turbo), and GPT-4 (OpenAI, 2023). Due to the cost of running these models, we generate outputs using unconstrained greedy decoding with a single generation per input document. For all our official experiments using InstructGPT, ChatGPT, GPT-4, we generated approximately 18 million tokens, 15 million tokens, 1 million tokens, respectively. All GPT experiments were conducted before December 2023.

**Settings**   We report results under two settings: predicted mentions, where only raw text is provided as input, and gold mentions, where the gold mention boundaries are provided as input. To obtain predicted mentions, we use the mentions output by dcoref as input into language model prompts.

---

[2]https://nlp.stanford.edu/software/dcoref.html

## 3.2 Results

| System | MUC | $B^3$ | $CEAF_4$ | CoNLL |
|---|---|---|---|---|
| | *Predicted mentions* | | | |
| *coref-mt5* | *87.8* | *82.6* | *79.5* | *83.3* |
| *coref-T0* | *87.6* | *82.4* | *79.5* | *83.2* |
| dcoref | 67.7 | 55.9 | 52.5 | 58.6 |
| weak-SpanBERT | 68.6 | 56.7 | **52.7** | 59.3 |
| Llama-2-Chat (70B) | 39.7 | 42.3 | 22.2 | 34.7 |
| CodeLlama (34B) | 57.5 | 40.6 | 25.3 | 41.1 |
| ChatGPT | 66.9 | 55.5 | 46.5 | 56.3 |
| InstructGPT | 70.4 | 58.4 | 51.7 | 60.1 |
| GPT-4 | **73.7** | **62.7** | 52.3 | **62.9** |
| | *Gold mentions* | | | |
| dcoref | 81.6 | 70.0 | 67.3 | 72.9 |
| Llama-2-Chat (7B) | 19.7 | 40.2 | 22.8 | 27.6 |
| Llama-2-Chat (70B) | 58.2 | 65.7 | 34.4 | 52.8 |
| CodeLlama (7B) | 71.5 | 54.5 | 31.1 | 52.4 |
| CodeLlama (34B) | 75.6 | 66.5 | 43.1 | 61.7 |
| ChatGPT | 86.2 | 79.3 | 68.3 | 77.9 |
| InstructGPT | 89.2 | 79.4 | 73.7 | 80.8 |
| GPT-4 | **93.7** | **88.8** | **82.8** | **88.4** |

Table 1: Result on English OntoNotes test set for predicted (top) and gold mentions (bottom). Fully supervised systems are *italicized*. The improvements of InstructGPT and GPT-4 over dcoref are statistical significant with $p < 0.05$, under the paired bootstrap resample test (Koehn, 2004).

**LLM-based coreference outperforms previous unsupervised systems**   Table 1 shows the results between different coreference systems. We note that prompting InstructGPT and GPT-4 outperforms dcoref for predicted mentions, with performance gaps increasing for gold mentions. However, this approach still considerably underperforms fully supervised systems. While all Llama-2 model variants underperform dcoref baseline, we note that CodeLlama significantly outperforms Llama-2-Chat. CodeLlama 7B even matches the performance of Llama-2-Chat 70B.

To further understand the strengths and weaknesses of instruction-tuned LMs for coreference, we break down the results according to different *resolution classes* (Lu & Ng, 2020). Specifically, for each coarse-grained mention class (named entity, pronoun, nominal), we compute the *resolution accuracy*, which is the percentage of anaphors correctly linked to an antecedent (Figure 2). We observe that InstructGPT does particularly well in pronoun resolution, corroborating previous work (Agrawal et al., 2022). It struggles more for named entities and the difficult nominal resolution. However, InstructGPT still remains competitive with dcoref for these classes, with the gaps increasing when gold mentions are provided. In particular, InstructGPT (and CodeLlama in gold mention setting) outperforms dcoref on challenging nominal phrases (Figure 2).

**A simple yet effective approach for supervised fine-tuning coreference with Llama-2**   To fairly compare our approach with supervised coreference models, we fine-tuned Llama-2 7B, 13B, and 70B using the full OntoNotes train set. The models are finetuned to generate the output document marked with coreference cluster IDs, given the document inputs formatted using the Document template. Gold mentions are provided during both training and testing. To enable efficient fine-tuning, we used LoRA (Hu et al., 2022) integrated with the HuggingFace library (Wolf et al., 2019). The largest model, Llama-2

| System | CoNLL $F_1$ |
|---|---|
| coref-T0 | **94.8** |
| SpanBERT+e2e | 91.1 |
| Llama-2 (7B) | 91.2 |
| Llama-2 (13B) | 92.8 |
| Llama-2 (70B) | 93.6 |

Table 2: Finetuning result on English OntoNotes dev set.

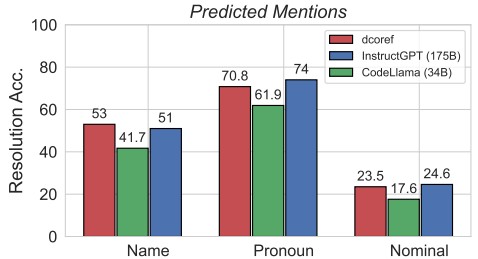
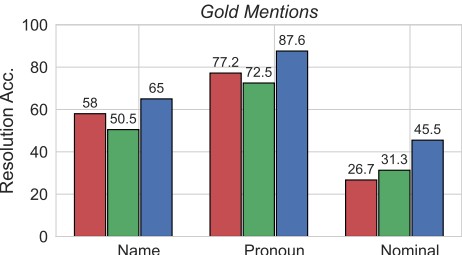

Figure 2: Resolution accuracy categorized by mention types (amongst the recalled mentions)

70B, was trained using 16 A40 GPUs over two days (five training epochs).

Table 2 compares two finetuned `Llama-2` models with two aforementioned supervised systems, `coref-T0` 11B parameters and `SpanBERT+e2e`. We note that finetuned `Llama-2` achieves competitive results in this setting, surpassing `SpanBERT+e2e` and approaching `coref-T0` despite having simpler text formats and generation procedures (e.g., no constrained beam search, no task-specific decoding actions). This indicates the feasibility of fine-tuning coreference with `Llama-2` using our simple prompt template.

### 3.3   The Importance of Mention Detection

While prompting LMs can be competitive with previous coreference systems, the quality of candidate mentions has a considerable effect on the final performance. We quantify the importance of high-quality Mention Detection (MD) by measuring the models' performance when inputting candidate mention sets generated by different mention detectors. Furthermore, we analyze the performance of prompting LMs for mentions with a simple template that outputs a list of named entities, pronouns, and nominal phrases, given an input text. We discuss these results below.

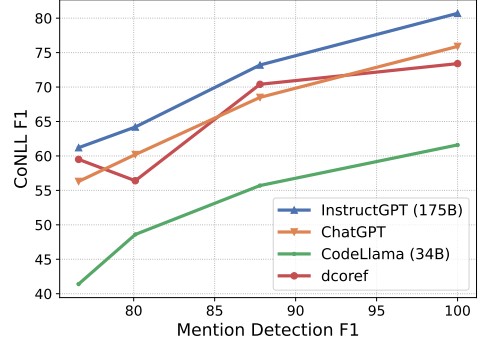

| Type | InstrGPT | GPT-4 | dcoref |
|------|----------|-------|--------|
| Name | 50.0 | 56.4 | **78.7** |
| Pronoun | 75.9 | 91.5 | **94.7** |
| Nominal | 18.7 | 19.8 | **52.7** |
| Overall | 51.5 | 59.9 | **77.5** |

Figure 3: (Left) CoNLL $F_1$ as a function of MD $F_1$, on OntoNotes dev set. All models were fed the same outputs from mention detection systems detailed in §A.3. (Right) Mention detection recall broken down by mention types. In addition to being overall worse than `dcoref`, `InstructGPT` and `GPT-4` particularly struggle with recalling nominal noun phrases.

**InstructGPT consistently outperforms dcoref as MD performance increases (Figure 3, left).** In general, coreference performances of all models improve as mention detection score increases. This is not surprising, as it has been similarly reported in previous work studying mention detection of neural coreference resolution systems (Lu & Ng, 2020). We further observe that `CodeLlama` underperforms while `ChatGPT` performs comparable to `dcoref` baseline. Nonetheless, we note that `InstructGPT` again consistently outperforms `dcoref`, regardless of MD performance.

| | |
|---|---|
| Mention Detection: | [Nine years] ago today, allegations of infidelity almost derailed [Bill Clinton]'s journey from hope to the White House. On [January 1992], [Gennifer Flowers] claims [she] had a 12 - year affair with [Bill Clinton]. Flowers went on "[Larry King] Live" in 1998 at the height of the [impeachment proceedings] against Mr. Clinton. [She] said [she] felt vindicated when [he] admitted under oath that [he]'d had an affair with [her] after denying [it] for years. |
| Antecedent Linking: (Gold Mentions) | Nine years ago today, [allegations of infidelity]$_1$ almost derailed [Bill Clinton's]$_2$ journey from hope to the White House. On January 1992, [Gennifer Flowers]$_3$ [claims]$_1$ [she]$_3$ had a 12 - year affair with [Bill Clinton]$_2$. [Flowers]$_4$ went on "Larry King Live" in 1998 at the height of the impeachment proceedings against [Mr. Clinton]$_2$. [She]$_3$ said [she]$_3$ felt vindicated when [he]$_2$ admitted under oath that [he]$_2$'d had [an affair with [her]$_3$]$_1$ after denying [it]$_1$ for years. |

Table 3: Qualitative examples of `InstructGPT` mention detection (top) and coreference resolution when gold mentions are given (bottom). Spans predicted by the model are wrapped around square brackets; Blue and red denote incorrect and correct predictions, respectively. **Mention Detection:** `InstructGPT` can predict most of the named entities and pronouns, but it still made numerous errors including extra entities (*Nine years*, *January 1992*), span errors (*Bill Clinton* vs *Bill Clinton's*), and missing mentions (*Mr. Clinton*). **Antecedent Linking:** `InstructGPT` exhibits near perfect antecedent linking ability, with the only exception being incorrectly linking *an affair with her* to *allegations of infidelity* (i.e. conflated entities error). Notably, it correctly resolved challenging cases like linking *claims* to *allegations of infidelity*. `InstructGPT` also exhibits some evidence of long-range ability when correctly resolving *it* to *allegations of infidelity*.

**Instruction-tuned LMs struggle with generating candidate mentions.** Figure 3 (right) shows that `InstructGPT` and `GPT-4` perform much worse than `dcoref`. Further analysis by mention types shows they particularly struggle to recall nominal mentions. A qualitative example in Table 3 demonstrates that while `InstructGPT` was able to recover a considerable portion of named entities and pronouns, it also made numerous errors, including span errors, extra entities, and missing mentions (Kummerfeld & Klein, 2013).

Given that what constitutes a mention can depend heavily on the annotation guidelines of specific datasets and domains, it may be challenging to ask a MD system to predict mentions without any labeled examples. Since Mention Detection plays a crucial role in coreference resolution (Wu & Gardner, 2021) as well as its generalizability to different domains, high-quality mention detection appears to be a pre-requisite for prompt-based coreference resolution. Fortunately, however, mention annotation has been shown to be much less costly than annotating full coreference chains (Gandhi et al., 2022).

## 4 Generalization Beyond OntoNotes

Although supervised neural models achieve superior results for coreference, they are also known to struggle when generalizing across domains, sometimes even underperforming rule-based systems (Moosavi & Strube, 2017). As such, recent research in coreference largely focus on the generalization ability of neural models beyond the OntoNotes dataset (Xia & Van Durme, 2021; Gandhi et al., 2022). Given that large LMs are pre-trained on lots of general-purpose data and are not optimized for a single coreference dataset, it seems plausible that instruction-tuned language models might also be effective across diverse texts. To explore this question, we examine how well instruction-tuned LMs generalize to different domains (§4.1), languages (§4.2), and time periods (§4.3). We mainly report results for `InstructGPT` and `ChatGPT`, given its competitive performance on OntoNotes while being less expensive than `GPT-4` (§3). The diverse coreference datasets considered in this analysis are given in Table 12. Since mention detection has been shown to be fairly challenging (§3.3), we evaluate the experiments in this section using gold mentions.

| Model | # Train Docs | $ON^{en}$ | LB | CI | WC | QBC | Avg. |
|---|---|---|---|---|---|---|---|
| TRANSFER-ON (Xia & Van Durme, 2021) | 2.8k → 10 | - | **85.0** | - | - | **85.0** | **85.0** |
| SpanBERT (Xia & Van Durme, 2021) | 0 → 10 | - | 69.0 | - | - | 65.0 | 67.0 |
| dcoref (Lee et al., 2013) | 0 → 0 | 72.9 | 55.4 | - | 72.4 | 34.8 | 59.0 |
| longdoc-PC (Toshniwal et al., 2021) | 36k → 0 | 76.8 | 81.1 | 66.5 | 67.0 | 77.3 | 73.7 |
| CodeLlama (34B) | 0 → 0 | 61.7 | 47.8 | 58.3 | 67.9 | 58.8 | 58.9 |
| InstructGPT | - | **80.8** | 77.0 | **72.6** | **72.9** | 68.3 | 74.3 |
| ChatGPT | - | 77.9 | 70.8 | 67.2 | 70.8 | 69.9 | 71.3 |

Table 4: CoNLL $F_1$ on different English coreference datasets, with the macro average shown in the last column. Best result is in **bold** while the second best is underlined. # train docs column indicates the number of train documents from the source domain → number of train documents from target domains. TRANSFER-ON and longdoc-PC were trained on large corpus of source examples; TRANSFER-ON and SpanBERT were fine-tuned on limited target examples; dcoref was not trained on any corpus. Overall, InstructGPT exhibits strong generalization results when using out-of-the-box.

## 4.1 Can LLMs resolve coreference across domains?

To study the robustness of our approach across domains, we use the datasets benchmarked in Toshniwal et al. (2021) due to the diversity in genres (news, Wikipedia, conversations), document lengths (long vs. short), and annotation guidelines (singletons vs. non-singletons). For evaluation, we follow the annotation schema of the corresponding dataset (i.e., if the dataset contains singletons, then we also output singletons). Similar to previous work in coreference domain adaptation (Xia & Van Durme, 2021; Toshniwal et al., 2021), we explore different systems where different types of source and target training data are available. Specifically, in addition to dcoref as in §3, we include the *trained models* TRANSFER-ON (Xia & Van Durme, 2021) and longdoc-PC (Toshniwal et al., 2021), which were respectively trained on the train set of OntoNotes$^{en}$ (2,802 annotated documents of newswire and religious texts) and PreCo (36,120 documents of reading comprehension examinations, collected in Chen et al. (2018)). TRANSFER-ON was then further finetuned on 10 labeled documents from the target domains. Additionally, we include the *pretrained encoder* SpanBERT (Xia & Van Durme, 2021) as a fine-tuning baseline (on a small amount of annotated data), where a pretrained SpanBERT encoder was not trained on a large source corpus and instead directly finetuned on 10 target documents. [3]

**InstructGPT is robust for coreference domain adapation.** Table 4 shows the coreference domain generalization for various systems. While InstructGPT is competitive with longdoc-PC, it still trails behind TRANSFER-ON considerably. This indicates that transfer learning is still a preferred method for coreference domain adaptation, particularly when a large corpus of training data and a few annotated documents in the target domain are available. On the other hand, when compared to models that were not trained on source coreference datasets such as dcoref and SpanBERT, InstructGPT outperforms them by a significant margin. This demonstrates the robustness of InstructGPT for coreference domain adaptation when using as a black-box model.

## 4.2 Can LMs also generalize coreference across languages?

To test the generalization of InstructGPT on resolving coreference across multiple languages, we experimented with Chinese and Arabic portions of OntoNotes and the multilingual coreference SemEval-2010 dataset (Recasens et al., 2010). A notable difference between OntoNotes and SemEval-2010 is the annotations of singletons, which has led to different evaluation methods for SemEval-2010. We follow the evaluation setting of previous work for each of the evaluated languages: excluding singletons from both predicted and evaluation clusters for Chinese and Arabic, while excluding singletons from predicted set but keeping

---

[3]Figure 1 of Xia & Van Durme (2021). Models summary detailed in Table 13.

them in evaluation sets for other languages. We refer to Section 5 of Bohnet et al. (2022) for more discussion on this.

Similar to §4.1, we compare `InstructGPT` with neural transfer-learning models from Xia & Van Durme (2021), `TRANSFER-EN` and `XLM-R`. Both use a pretrained `XLM-RoBERTa-large` encoder fine-tuned with 10 documents from the target language. We note that `TRANSFER-EN` was previously trained on English OntoNotes before continuing training on the target language, which makes it a stronger model than `XLM-R`. `TRANSFER-EN` and `XLM-R` correspond to `TRANSFER-ON` and `SpanBERT` from §4.1, respectively, with the only difference being the pretrained encoder (XLM-R vs. SpanBERT).

**InstructGPT can also effectively resolve coreference across languages.** From Table 5, we observe similar conclusions to §4.1: continued learning using a large source corpus with a handful of annotated examples from target languages still performs the best. Nonetheless, `InstructGPT` was able to outperform `XLM-R` across all languages, and is even on par with `TRANSFER-EN` for Chinese and Dutch. This result indicates the importance of a source English coreference corpus for continued learning.

| Lang. | TRFER-EN 2.8k→ 10 | XLM-R 0 → 10 | InstrGPT |
|---|---|---|---|
| Chinese (zh) | 75.0 | 70.0 | **77.3** |
| Arabic (ar) | **80.0** | 49.0 | 65.6 |
| Catalan (ca) | **52.0** | 29.0 | 41.9 |
| Dutch (nl) | **71.0** | 42.0 | 70.8 |
| Italian (it) | **46.0** | 25.0 | 41.4 |
| Spanish (es) | **57.0** | 35.0 | 42.2 |

Table 5: CoNLL $F_1$ on the Chinese and Arabic portions of OntoNotes and SemEval-2010 dataset. Best result is in **bold** while the second best is underlined.

## 4.3 What about different time periods?

An interesting dimension to analyze the robustness of coreference generalization is temporal changes (Agarwal & Nenkova, 2022; Liu & Ritter, 2023), since having coreference systems that can generalize beyond datasets that were created over a decade ago (e.g., OntoNotes) can be beneficial. To that end, we compare `dcoref` and several instruction-tuned LMs on three new silver-annotated coreference datasets from different time periods: **WSJ-1989**, **WSJ-2019**, and **WSJ-2023**, each containing 56 Wall Street Journal articles from 1989, 2015-2019, and 2023, respectively. `WSJ-1989` is a subset of the OntoNotes dev set and thus contains gold coreference annotation. `WSJ-2019` was sampled from the RealNews dataset (Zellers et al., 2019) dated from February 2015 to February 2019, and `WSJ-2023` from the WSJ website between May and June 2023. Since these two datasets do not have coreference annotations, we used SpanBERT (Joshi et al., 2020), which was fine-tuned on the in-domain OntoNotes train set, to obtain *silver annotations* for all three datasets. We then evaluate the models on these silver annotations, with mentions given as before. Further details on how we sampled and annotated these datasets are presented in §A.4.

| Model | 1989 (G) | 1989 (S) | 2019 (S) | 2023 (S) | $\sigma^2$ |
|---|---|---|---|---|---|
| dcoref | 72.4 | 70.8 | 63.6 | 66.9 | 15.7 |
| CodeLlama-34B | 61.9 | 57.4 | 55.7 | 55.3 | 9.1 |
| InstructGPT | **80.9** | **78.2** | **80.5** | **81.7** | **2.3** |
| ChatGPT | 76.8 | 75.3 | 76.7 | 74.3 | 2.5 |

Table 6: CoNLL $F_1$ and variance (last column) on Wall Street Journal articles from different time periods. G and S denote Gold and Silver annotations, respectively. Prompting LMs appears more robust to temporal changes than `dcoref`.

**Prompting instruction-tuned LMs is robust to temporal changes.** Table 6 shows the results. We first observe a decrease when moving from gold to silver annotations for all models. More importantly, we see more degradation and variance in performance of `dcoref` for the different temporal datasets, whereas the variance is less pronounced for `InstructGPT`

and `ChatGPT`. While `CodeLlama-34B` underperforms `dcoref` baseline, it also observes less variance when evaluated on different temporal datasets.

## 5 Related Work

**Domain Adaptation for Coreference**   Previous work has reported that neural models trained on a single dataset struggled with out-of-domain generalization, with some performing worse than rule-based systems (Moosavi & Strube, 2017). Several solutions to this challenge have been proposed with varying success: Xia & Van Durme (2021) shows that continued training can help generalize to different domains and languages with as few as 10 annotated documents, and Toshniwal et al. (2021) leverages joint training on large coreference corpora with different annotations to help neural models adapt to new domains. Recently, Gandhi et al. (2022) demonstrates that adapting mention annotations to new domains instead of the entire coreference chains is more cost-efficient while also improving domain adaptation performance. In contrast to the above work, we propose to prompt general-purpose language models for coreference resolution and show promising generalization capabilities across domains. Our findings also align with contemporaneous work Nori et al. (2023), which shows that prompting can unlock specialized capabilities in general-purpose LLMs.

**Conditional Text Generation for Coreference**   Research in coreference resolution has been dominated by neural span-based models that score coreference links between spans (Lee et al., 2017; Joshi et al., 2020). Recently, a new paradigm for coreference starts to emerge: formulating coreference resolution as conditional text generation (Liu et al., 2022; Bohnet et al., 2022; Zhang et al., 2023). Both Liu et al. (2022) and Bohnet et al. (2022) fine-tuned T5-based models on sequences of structured-building actions, with the former achieving competitive results for structured prediction tasks and the latter achieving SOTA results for coreference resolution. Zhang et al. (2023) finetuned T0 models on a simpler text sequences that directly encode coreference annotations, yet achieved comparable results to Bohnet et al. (2022). While our work falls into this category, we are interested the intrinsic ability of the language model to resolve coreference, using an autoregressive language model on an instruction-based prompt format. Another coreference annotation framework that is similar to ours is TANL (Paolini et al., 2021), which also formulates structured prediction as a text generation task. The main difference between our work and TANL is that TANL is evaluated in fully supervised setting using multi-task learning, whereas we evaluated instruction-tuned LLMs without explicit fine-tuning on the CoNLL train set.

**Prompting LMs for Coreference**   With the success of zero-shot and few-shot prompting of large language models on various NLP benchmarks, we ask to what extent this success translates to more traditional NLP tasks like coreference resolution. Manning et al. (2020) shows evidence of linguistic abilities in masked LMs, and Blevins et al. (2022) presents a structured prompting approach that achieves strong few-shot results for sequence tagging tasks. For coreference resolution, prior work has mostly focused on few-shot learning for sentence-level, syntactically simple coreference datasets such as Winograd Schema Challenge (Levesque et al., 2012) and for pronoun resolution on clinical data (Agrawal et al., 2022).

## 6 Conclusion

In this paper, we study how well instruction-tuned language models resolve coreference via prompting. We demonstrate the feasibility of this approach on the CoNLL-2012 benchmark, surpassing previous unsupervised systems but still underperforming state-of-the-art supervised models. Interestingly, prompting instruction-tuned LMs appears to generalize well across a wide range of domains, languages, and time periods, particularly if no training examples are given. Nonetheless, it still trails behind continued learning with a large training corpus in the source domain and a handful of annotated examples in the target domain.

Even with the surprising effectiveness of language model technologies in various applications, coreference resolution remains a challenging task. In addition, it continues to be an important subtasks of the Information Extraction (IE) pipeline (particularly for document-level IE), with usages in applications such as extracting scientific documents, answering queries that require aggregation over a collection of unstructured documents, etc. Given the emerging effectiveness of LMs on a variety of NLP tasks and the importance of adapting coreference models to different domains, we hope this study can help shed some light on future research in both directions.

## Limitations

Because OpenAI GPT models are proprietary models, we do not know whether or not OntoNotes was included in its training data. However, at the time of writing, there is some evidence against OntoNotes data contamination. First, a previous probe that aimes to measure data contamination and memorization of OntoNotes on ChatGPT showed negative results. [4] Second, our experiment in §4.3 includes data sampled after the models' training cutoff date (September 2021), yet still shows a robust $F_1$. Finally, the conclusions in this paper still stand regardless of whether or not these models trained on OntoNotes: (1) prompting instruction-tuned LMs is a feasible strategy for coreference resolution, and (2) although this approach has unique strengths and weaknesses, it is robust across many domains, languages, and time periods.

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

# A Appendix

## A.1 Preliminaries on Prompt Formatting

| Prompt Format | GPT-J | InstructGPT | GPT-4 |
|---|---|---|---|
| QA 0-shot | 4.2 | 22.9 | 15.3 |
| QA $k$-shot | 50.2 | 61.2 | 67.3 |
| Doc 0-shot | 24.2 | **81.7** | **86.2** |
| Doc $k$-shot | **58.2** | 65.4 | 84.0 |

Table 7: Results of different prompt configurations for coreference on a subset of OntoNotes dev set, using gold mentions. Note that dcoref achieves 71.9 $F_1$ on the same dataset.

**Question-Answer Prompting for Coreference**  During preliminary studies, we experimented with different approaches for prompting coreference from previous work (Agrawal et al., 2022; Ouyang et al., 2022). However, we found that the common Question-Answer template performed consistently worse than the deterministic coreference system dcoref (Lee et al., 2013), despite adding in-context demonstrations to provide formatting guidance (Agrawal et al., 2022). Qualitative, while this format seems effective at resolving pronouns, it struggles with more ambiguous nominal noun phrases. For example, asking it to resolve *an affair with her* in Table 15 using QA template would yield an incorrect answer *allegations of infidelity*.

**Question-Answer vs. Document Template**  We further found that the Document template was more effective than the QA template at resolving coreference. Table A.1 shows the results on several LMs and prompt configurations. For $k$-shot experiments, we first randomly sampled a set of 64 documents from the OntoNotes train set. For each development example, we again randomly sampled in-context demonstrations from this smaller train set until the max context len is exceeded (average 5 demonstrations for QA and 2 for Doc). We observe that larger LMs such as InstructGPT outperformed dcoref using Document template. Interestingly, adding in-context demonstrations for this approach did not improve the LMs performance. We hypothesize that the Document prompts need less formatting guidance in the answer compared to open-ended QA, hence in-context demonstrations would be less effective here. We further note that this template is loosely similar to the entity-based approach to coreference, where the model links a mention with previous clusters, as opposed to the mention-paired approach exemplified by the QA template (Jurafsky & Martin, 2000). In addition, extracting the predicted clusters from the generated text is easier than other formats, as InstructGPT would directly annotate the text with the cluster information (we extract cluster information using a simple fuzzy string matching algorithm by comparing the output text to input text, sentence-by-sentence).

## A.2 CODI-CRAC 2022 Experiments

We compare our results with a contemporary work of Gan et al. (2024), which also evaluated the capabilities of different LMs for coreference resolution. Unlike our work, however, Gan et al. (2024) mainly experimented with the CODI-CRAC 2022 dataset (Yu et al., 2022). For comparison, we use our Document template prompt with GPT-4 and Llama-2-70B on the LIGHT partition of the dataset. Results on CoNLL $F_1$ are reported in Table 8. We observe the superior performance of our prompting technique over Gan et al. (2024) models. Strong results on recent coreference dataset CODI-CRAC 2022 also further supported LMs zero-shot coreference generalization ability.

## A.3 Mention Detection Experiments

To experiment with different qualities of candidate mention sets, we adapting different existing methods for the task of Mention Detection: given an input document, extract all the candidate mentions from the text. For mention detection, we mainly consider the mention

| | CoNLL $F_1$ |
|---|---|
| Llama-2-70B (Gan et al., 2024) | 36.1 |
| GPT-4 (Gan et al., 2024) | 51.4 |
| Llama-2-70B (Ours) | 46.3 |
| GPT-4 (Ours) | 84.7 |

Table 8: CODI-CRAC 2022 results.

detector from `dcoref` as well as the prompting of `InstructGPT` for MD using template in Table 11. In addition, to see the effects of having high-quality mentions on `dcoref` and `InstructGPT`, we also consider outputs from `SpanBERT-large` trained on OntoNotes train set (Joshi et al., 2020) and a NER tagger with `xlm-roberta-large` (Conneau et al., 2020) trained on BIO labels adapted from OntoNotes annotations. We note that these systems are not directly comparable to each other, since they were trained on different annotatations: `SpanBERT-large` on full coreference data and `xlm-roberta-large` on non-nested MD data.

| | Train | P | R | $F_1$ |
|---|---|---|---|---|
| `SpanBERT-large` | CR | 89.1 | 86.6 | 87.8 |
| `xlm-roberta-large` | MD | 83.3 | 76.3 | 80.1 |
| `dcoref` | ∅ | 75.8 | 77.4 | 76.6 |
| `InstructGPT` | - | 42.1 | 51.8 | 46.5 |

Table 9: MD results of different systems considered in Figure 3. `SpanBERT-large` was trained on full coreference (CR) data, `xlm-roberta-large` trained on mention-annotated-only (MD) OntoNotes train set, `dcoref` was not trained on any corpus, and `InstructGPT` exact training procedures are unknown.

## A.4 Temporal Generalization for Coreference

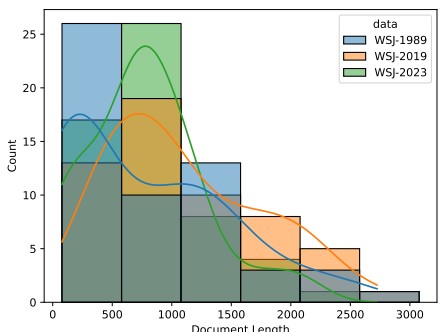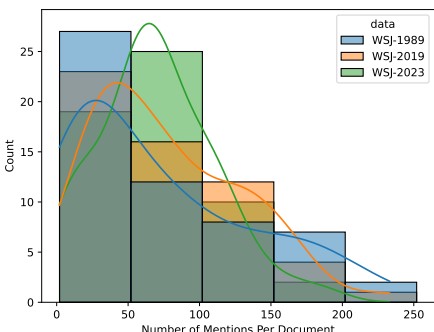

Figure 4: Distributions of `WSJ-1989` (blue), `WSJ-2019` (orange), and `WSJ-2023` (green) based on document length (left) and number of mentions per document (right). The number of mentions per document is measured using the silver annotations from SpanBERT (Joshi et al., 2020).

**Data Sampling** To sample the appropriate data for this experiment, we start with the Wall Street Journal sections of the RealNews (Zellers et al., 2019) and OntoNotes dev set. We used SpanBERT (Joshi et al., 2020) to label all 56 WSJ articles from OntoNotes to obtain `WSJ-1989` (CoNLL $F_1$ using SpanBERT on `WSJ-1989` is shown on Table 10). To create `WSJ-2019`, we first labeled all 191 WSJ articles from RealNews using SpanBERT as above. We then sampled 56 articles using stratified sampling based on two features: document length and number of mentions per document. Specifically, we partitioned the WSJ RealNews articles into bins

based on document lengths (bin size = 500 tokens), and for each document-length bin we further partitioned based on the number of mentions (mention size = 50). We then sampled the appropriate number of documents (i.e., the number of WSJ-1989 documents in each partition) for each bin to obtain WSJ-2019. For WSJ-2023, we randomly collected 56 articles from the WSJ website dated between May and June 2023 based on document lengths and topics. The distributions of three datasets are shown in Figure 4.

| Dataset | CoNLL $F_1$ |
|---------|-------------|
| OntoNotes | 79.2 |
| WSJ-1989 | 74.5 |

Table 10: CoNLL $F_1$ when running SpanBERT (Joshi et al., 2020) on OntoNotes dev set and WSJ-1989.

---

**Question-Answer Template**

```
Instructions: Please carefully read the following passages. For each passage,
you must identify which noun the mention marked in *bold* refers to.
Context: In the summer of 2005, a picture that people have long been looking
forward to started emerging with frequency in various major Hong Kong media.
With their unique charm, these well-known cartoon images once again caused
Hong Kong to be a focus of worldwide attention. The world's fifth Disney park
 will soon open to the public here. The most important thing about Disney
is that *it* is a global brand.
Question: What does *it* refer to?
Answer: *it* refers to Disney.
```

**Document Template**

```
Annotate all entity mentions in the following text with coreference clusters.
Use Markdown tags to indicate clusters in the output, with the following format
[mention](#cluster_name)
Input: In the summer of 2005, a picture that people have long been looking forward
to started emerging with frequency in various major [Hong Kong](#) media.
With [their](#) unique charm, [these well-known cartoon images](#) once again
caused [Hong Kong](#) to be a focus of worldwide attention. [The world's fifth
[Disney](#) park](#) will soon open to the public here. The most important thing
about [Disney](#) is that [it](#) is a global brand.
Output: In the summer of 2005, a picture that people have long been looking forward
to started emerging with frequency in various major [Hong Kong](#cluster_0) media.
With [their](#cluster_1) unique charm, [these well-known cartoon images](#cluster_1)
once again caused [Hong Kong](#cluster_0) to be a focus of worldwide attention.
[The world's fifth [Disney](#cluster_3) park](#cluster_2) will soon open to the public
here. The most important thing about [Disney](#cluster_3) is that [it](#cluster_3)
is a global brand.
```

**Mention Detection Template**

```
In the following text, list all named entities, pronouns, and nominal noun phrases
according to the OntoNotes conventions.
Input: In the summer of 2005, a picture that people have long been looking forward
to started emerging with frequency in various major Hong Kong media. With their
unique charm, these well-known cartoon images once again caused Hong Kong to be
a focus of worldwide attention. The world's fifth Disney park will soon open to
the public here. The most important thing about Disney is that it is a global brand.
Output:
Named Entities: Hong Kong
Pronouns: their, it, many, its, that, its, this
Nominal Noun Phrases: these well-known cartoon images, the world's fifth Disney park
```

Table 11: Examples of coreference and mention detection prompt templates used in this work.

| Dataset | Lang. | Train | Dev | Test | Toks/Doc (Test) | % Singletons | Domains |
|---|---|---|---|---|---|---|---|
| OntoNotes[en] | English | 2802 | 343 | 348 | 489 | 0.0 | News, magazine, transcripts, biblical text |
| Litbank | English | 80 | 10 | 10 | 2105 | 19.8 | Literature (Project Gutenberg) |
| Character Iden. | English | 987 | 122 | 192 | 262 | 6.4 | Movie conversations |
| WikiCoref | English | 0 | 0 | 30 | 1996 | 0.0 | Wikipedia |
| QuizBowlCoref | English | 0 | 0 | 400 | 126 | 26.0 | Trivia questions |
| OntoNotes[zh] | Chinese | 1729 | 254 | 218 | 412 | 0.0 | News, magazine |
| OntoNotes[ar] | Arabic | 359 | 44 | 44 | 681 | 0.0 | News |
| SemEval[ca] | Catalan | 829 | 142 | 167 | 293 | 45.9 | News |
| SemEval[nl] | Dutch | 145 | 23 | 72 | 666 | 13.0 | Magazine |
| SemEval[it] | Italian | 80 | 18 | 46 | 891 | 61.9 | Wikipedia, blogs, news, dialogues |
| SemEval[es] | Spanish | 875 | 140 | 168 | 303 | 47.7 | News |
| WSJ-1989 | English | 0 | 0 | 56 | 632 | 0.0 | News (Wall Street Journal articles) |
| WSJ-2019 | English | 0 | 0 | 56 | 858 | 0.0 | News (Wall Street Journal articles) |
| WSJ-2023 | English | 0 | 0 | 56 | 688 | 0.0 | News (Wall Street Journal articles) |

Table 12: Detailed statistics of datasets. Following prior work on multilingual coreference resolution (Bohnet et al., 2022; Xia & Van Durme, 2021), we excluded SemEval English as the data overlaps with English OntoNotes, and SemEval-2010 German due to licensing issues. We also excluded GAP, WSC, and PreCo from the benchmarks in Toshniwal et al. (2021): GAP and WSC due to the simplicity of these datasets as well as being extensively studied by previous work, and PreCo for not being able to obtain it despite contacting the authors.

| Model | Prior Work | Description |
|---|---|---|
| InstructGPT | Ouyang et al. (2022) | pretrained on massive amount of data |
| dcoref | Lee et al. (2013) | deterministic system developed on OntoNotes[en]; 0-shot on target data |
| longdoc-PC | Toshniwal et al. (2021) | joint training; 0-shot on target data |
| TRANSFER-ON | Xia & Van Durme (2021) | trained on OntoNotes[en]; few-shot on target data |
| SpanBERT | Xia & Van Durme (2021) | pretrained on unlabeled corpus; few-shot on target data |
| TRANSFER-EN | Xia & Van Durme (2021) | trained on OntoNotes[en]; few-shot on target data |
| XLM-R | Xia & Van Durme (2021) | pretrained on unlabeled corpus; few-shot on target data |

Table 13: Summary of models

| System | MUC | | | $B^3$ | | | $CEAF_{\phi_4}$ | | | CoNLL |
|---|---|---|---|---|---|---|---|---|---|---|
| | P | R | $F_1$ | P | R | $F_1$ | P | R | $F_1$ | $F_1$ |
| *Predicted mentions* | | | | | | | | | | |
| *coref-mt5* (Bohnet et al., 2022) | 87.4 | 88.3 | 87.8 | 81.8 | 83.4 | 82.6 | 79.1 | 79.9 | 79.5 | 83.3 |
| *SpanBERT+e2e* (Joshi et al., 2020) | 85.8 | 84.8 | 85.3 | 78.3 | 77.9 | 78.1 | 76.4 | 74.2 | 75.3 | 79.6 |
| dcoref (Lee et al., 2013) | 67.7 | 67.8 | 67.7 | 59.3 | 52.8 | 55.9 | 49.3 | 56.0 | 52.5 | 58.6 |
| weak-SpanBERT (Stolfo et al., 2022) | 67.4 | 69.8 | 68.6 | 52.4 | 61.8 | 56.7 | 54.1 | 51.4 | **52.7** | 59.3 |
| llama-2-70B-chat (Touvron et al., 2023) | 60.2 | 29.6 | 39.7 | 55.8 | 34.0 | 42.3 | 14.7 | 45.5 | 22.2 | 34.7 |
| codellama-34B (Rozière et al., 2023) | 54.3 | 61.0 | 57.5 | 34.3 | 49.6 | 40.6 | 22.4 | 29.1 | 25.3 | 41.1 |
| InstructGPT (Ouyang et al., 2022) | 71.1 | 69.7 | 70.4 | 58.1 | 58.6 | 58.4 | 60.6 | 45.1 | 51.7 | 60.1 |
| ChatGPT (OpenAI, 2022) | 67.3 | 66.5 | 66.9 | 54.3 | 56.8 | 55.5 | 43.9 | 49.5 | 46.5 | 56.3 |
| gpt-4 (OpenAI, 2023) | 73.9 | 73.5 | **73.7** | 60.8 | 64.7 | **62.7** | 49.3 | 55.7 | 52.3 | **62.9** |
| *Gold mentions* | | | | | | | | | | |
| dcoref (Lee et al., 2013) | 90.0 | 74.5 | 81.6 | 84.2 | 59.7 | 70.0 | 74.4 | 61.4 | 67.3 | 72.9 |
| llama-2-7B-chat (Touvron et al., 2023) | 60.3 | 11.8 | 19.7 | 86.8 | 26.2 | 40.2 | 15.9 | 40.5 | 22.8 | 27.6 |
| llama-2-70B-chat (Touvron et al., 2023) | 86.7 | 43.8 | 58.2 | 88.8 | 52.2 | 65.5 | 24.0 | 60.3 | 34.4 | 52.8 |
| codellama-7B (Rozière et al., 2023) | 72.2 | 70.7 | 71.5 | 45.2 | 68.7 | 54.5 | 30.1 | 32.1 | 31.1 | 52.4 |
| codellama-34B (Rozière et al., 2023) | 78.5 | 72.9 | 75.6 | 63.5 | 69.9 | 66.5 | 39.0 | 48.3 | 43.1 | 61.7 |
| InstructGPT (Ouyang et al., 2022) | 89.6 | 88.9 | 89.2 | 76.0 | 89.2 | 79.4 | 84.8 | 65.2 | 73.7 | 80.8 |
| ChatGPT (OpenAI, 2022) | 88.2 | 84.4 | 86.2 | 79.3 | 79.3 | 79.3 | 65.6 | 71.2 | 68.3 | 77.9 |
| gpt-4 (OpenAI, 2023) | 93.8 | 93.7 | **93.7** | 86.5 | 91.1 | **88.8** | 83.5 | 82.0 | **82.8** | **88.4** |

Table 14: Result on English OntoNotes test set for predicted mentions (top) and gold mentions (bottom). Fully supervised systems are italicized.

| | |
|---|---|
| Mention Detection: (`InstructGPT`) | [Nine years] ago today, allegations of infidelity almost derailed [Bill Clinton]'s journey from hope to the White House. [Bob Glascoff] tracks the life of the "other woman" in [today's edition] of "Headliners." On [January 1992], [Gennifer Flowers] claims [she] had a 12 - year affair with [Bill Clinton]. Although Mr. Clinton denied having a relationship with Flowers, [he] did speak of bringing "pain" to [his] marriage during a [joint television interview] with [his] wife, Hillary. Flowers went on "[Larry King] Live" in 1998 at the height of the [impeachment proceedings] against Mr. Clinton. [She] said [she] felt vindicated when [he] admitted under oath that [he]'d had an affair with [her] after denying [it] for years. A [federal judge] recently dismissed a [defamation lawsuit] [she] brought against [Hillary Rodham Clinton] and two former presidential aides. With "Headliners," I'm [Bob Glascoff]. |
| Predicted Mentions: (`InstructGPT`) | Nine years ago today, allegations of infidelity almost derailed [Bill Clinton's]$_3$ journey from hope to the White House. Bob Glascoff tracks the life of the "other woman" in today's edition of "[Headliners]$_5$." On January 1992, [Gennifer Flowers]$_6$ claims [she]$_6$ had a 12-year affair with [Bill Clinton]$_3$. Although [Mr. Clinton]$_3$ denied having a relationship with [Flowers]$_6$, [he]$_3$ did speak of bringing "pain" to [his]$_3$ marriage during a joint television interview with [his]$_3$ wife, Hillary. [Flowers]$_6$ went on ["Larry King Live"]$_5$ in 1998 at the height of the impeachment proceedings against [Mr. Clinton]$_3$. [She]$_6$ said [she]$_6$ felt vindicated when [he]$_3$ admitted under oath that [he]$_3$'d had [an affair with [her]$_6$ ]$_6$ after denying [it]$_6$ for years. A federal judge recently dismissed a defamation lawsuit [she]$_6$ brought against Hillary Rodham Clinton and two former presidential aides. With "[Headliners]$_5$," I'm Bob Glascoff. |
| Gold Mentions: (`dcoref`) | Nine years ago [today]$_1$, allegations of infidelity almost derailed [Bill Clinton's]$_3$ journey from hope to the White House. Bob Glascoff tracks the life of the "other woman" in [today's]$_1$ edition of "[Headliners]$_5$." On January 1992, [Gennifer Flowers]$_6$ claims [she]$_6$ had a 12 - year affair with [Bill Clinton]$_3$. Although [Mr. Clinton]$_3$ denied having a relationship with [Flowers]$_6$, [he]$_3$ did speak of bringing "pain" to [his]$_3$ marriage during a joint television interview with [his]$_3$ wife, Hillary. [Flowers]$_6$ went on "Larry King Live" in 1998 at the height of the impeachment proceedings against [Mr. Clinton]$_3$. [She]$_6$ said [she]$_6$ felt vindicated when [he]$_3$ admitted had [an affair with [her]$_6$]$_8$ after denying [it]$_8$ for years. A federal judge recently dismissed a defamation lawsuit [she]$_6$ brought against Hillary Rodham Clinton and two former presidential aides. With "[Headliners]$_5$," [I]$_5$'m Bob Glascoff. |
| Gold Mentions: (`InstructGPT`) | Nine years ago [today]$_1$, [allegations of infidelity]$_2$ almost derailed [Bill Clinton's]$_3$ journey from hope to the White House. [Bob Glascoff]$_4$ tracks the life of [the "other woman"]$_6$ in [today's]$_1$ edition of "[Headliners]$_5$." On January 1992, [Gennifer Flowers]$_6$ [claims]$_2$ [she]$_6$ had a 12 - year affair with [Bill Clinton]$_3$. Although [Mr. Clinton]$_3$ denied having a relationship with [Flowers]$_6$, [he]$_3$ did speak of bringing "pain" to [his]$_3$ marriage during a joint television interview with [[his]$_3$ wife, Hillary]$_7$. [Flowers]$_6$ went on "Larry King Live" in 1998 at the height of the impeachment proceedings against [Mr. Clinton]$_3$. [She]$_6$ said [she]$_6$ felt vindicated when [he]$_3$ admitted under oath that [he]$_3$'d had [an affair with [her]$_6$]$_2$ after denying [it]$_2$ for years. A federal judge recently dismissed a defamation lawsuit [she]$_6$ brought against [Hillary Rodham Clinton]$_7$ and two former presidential aides. With "[Headliners]$_5$," [I]$_4$'m Bob Glascoff. |
| Gold Output: | Nine years ago [today]$_1$, [allegations of infidelity]$_2$ almost derailed [Bill Clinton's]$_3$ journey from hope to the White House. [Bob Glascoff]$_4$ tracks the life of [the "other woman"]$_6$ in [today's]$_1$ edition of "[Headliners]$_5$." On January 1992, [Gennifer Flowers]$_6$ [claims]$_2$ [she]$_6$ had a 12 - year affair with [Bill Clinton]$_3$. Although [Mr. Clinton]$_3$ denied having a relationship with [Flowers]$_6$, [he]$_3$ did speak of bringing "pain" to [his]$_3$ marriage during a joint television interview with [[his]$_3$ wife, Hillary]$_7$. [Flowers]$_6$ went on "Larry King Live" in 1998 at the height of the impeachment proceedings against [Mr. Clinton]$_3$. [She]$_6$ said [she]$_6$ felt vindicated when [he]$_3$ admitted under oath that [he]$_3$'d had [an affair with [her]$_6$]$_8$ after denying [it]$_8$ for years. A federal judge recently dismissed a defamation lawsuit [she]$_6$ brought against [Hillary Rodham Clinton]$_7$ and two former presidential aides. With "[Headliners]$_5$," [I]$_4$'m Bob Glascoff. |

Table 15: A qualitative examples of `InstructGPT` and `dcoref` coreference predictions under various setting: Row 1 shows `InstructGPT` mention detection result; Row 2 shows `InstructGPT` coreference results using `dcoref` predicted mentions; Row 3 and 4 show `dcoref` and `InstructGPT` coreference results using gold mentions; and last row is the gold output.

| | |
|---|---|
| Mention Detection: (InstructGPT) | [Mai Po Marshes] adjacent to [Wetland Park] is a [major wildlife habitat] within [Asia]. Each year, over 50,000 migratory birds fly over [Hong Kong]'s skyscrapers and choose to roost for winter here. As a result, [three different types of aviaries] were built in [[Hong Kong] [Wetland Park]]. These have become the best spots to observe birds. Among [common birds], a rather special one is the black-faced spoonbill. [It] is [an endangered bird species] throughout the [world]. Uh-huh. Ah, there are only about 1,500 in the [world]. Wow. Um, however, each year, about [two to three hundred] of [them] come to [Hong Kong] to spend the winter. Some of [them], er, have stayed in [[Hong Kong] [Wetland Park]]. Uh-huh. So, [our] park's logo is unique, featuring this black-faced spoonbill , [which] hopefully can draw [people's attention]. Uh-huh. |
| Gold Mentions: (dcoref) | Mai Po Marshes adjacent to $[\text{Wetland Park}]_0$ is a major wildlife habitat within Asia. Each year, over 50,000 migratory birds fly over $[\text{Hong Kong's}]_1$ skyscrapers and choose to roost for winter here. As a result, three different types of aviaries were built in $[\text{Hong Kong Wetland Park}]_0$. These have become the best spots to observe birds. Among common birds, $[\text{a rather special one}]_2$ is the black-faced spoonbill. $[\text{It}]_2$ is an endangered bird species throughout $[\text{the world}]_3$. Uh-huh. Ah, there are only about 1,500 in $[\text{the world}]_3$. Wow. Um, however, each year about two to three hundred of $[\text{them}]_4$ come to $[\text{Hong Kong}]_1$ to spend the winter. Some of $[\text{them}]_4$, er, have stayed in $[\text{Hong Kong Wetland Park}]_0$. Uh-huh. So, $[\text{our park's}]_0$ logo is unique, featuring this black-faced spoonbill, which hopefully can draw people's attention. Uh-huh. |
| Gold Mentions: (InstructGPT) | Mai Po Marshes adjacent to Wetland Park is a major wildlife habitat within Asia. Each year, over 50,000 migratory birds fly over $[\text{Hong Kong's}]_1$ skyscrapers and choose to roost for winter here. As a result, $[\text{three different types of aviaries}]_2$ were built in $[\text{Hong Kong Wetland Park}]_1$. $[\text{These}]_2$ have become the best spots to observe birds. Among common birds, $[\text{a rather special one}]_3$ is the black-faced spoonbill. $[\text{It}]_3$ is an endangered bird species throughout $[\text{the world}]_4$. Uh-huh. Ah, there are $[\text{only about 1,500 in } [\text{the world}]_4]_4$. Wow. Um, however, each year, $[\text{about two to three hundred of } [\text{them}]_3]_3$ come to $[\text{Hong Kong}]_1$ to spend the winter. Some of $[\text{them}]_3$, er, have stayed in $[\text{Hong Kong Wetland Park}]_1$. Uh-huh. So, $[\text{our park's}]_1$ logo is unique, featuring this black-faced spoonbill, which hopefully can draw people's attention. Uh-huh. |
| Gold Output: | Mai Po Marshes adjacent to $[\text{Wetland Park}]_2$ is a major wildlife habitat within Asia. Each year, over 50,000 migratory birds fly over $[\text{Hong Kong's}]_0$ skyscrapers and choose to roost for winter here. As a result, $[\text{three different types of aviaries}]_1$ were built in $[\text{Hong Kong Wetland Park}]_2$. $[\text{These}]_1$ have become the best spots to observe birds. Among common birds, $[\text{a rather special one}]_3$ is the black-faced spoonbill. $[\text{It}]_3$ is an endangered bird species throughout $[\text{the world}]_4$. Uh-huh. Ah, there are $[\text{only about 1,500 in } [\text{the world}]_4]_5$. Wow. Um, however, each year, $[\text{about two to three hundred of } [\text{them}]_5]_6$ come to $[\text{Hong Kong}]_0$ to spend the winter. Some of $[\text{them}]_6$, er, have stayed in $[\text{Hong Kong Wetland Park}]_2$. Uh-huh. So, $[\text{our park's}]_2$ logo is unique, featuring this black-faced spoonbill, which hopefully can draw people's attention. Uh-huh. |

Table 16: An example where InstructGPT struggles to resolve coreference, even on gold mentions. The most notable case is with nested mentions (e.g., $[\text{about two to three hundred of } [\text{them}]_3]_3$).

