# OpenReview forum: "Are Language Models Robust Coreference Resolvers?"
_colmweb.org/COLM/2024/Conference — COLM_

### Official Review · Reviewer_3vsC · 2024-05-08

**Rating:** 7
**Confidence:** 3
**Ethics Flag:** 1

**Summary:**

The paper explores entity coreference resolution, focusing on leveraging LLMs through prompting. Results indicate promising potential, although the method still falls behind supervised models and requires a robust mention detector. Nonetheless, the study demonstrates the generalization capabilities of LLMs across different contexts.

**Reasons To Accept:**

1. This paper delves into the exploration of various prompts in coreference resolution.
2. The experimental setup outlined in the paper is methodically constructed, ensuring reliability, and the conclusions drawn from the results are deemed acceptable.
3. The study thoroughly investigates the generalization capabilities of LLMs, providing valuable insights into their adaptability across diverse domains, languages, and time periods.

**Reasons To Reject:**

1. While OntoNotes has been a valuable resource for coreference research, its age may limit the generalizability of the findings. Consideration of experiments on newer datasets with more complex mentions and coreference examples would enhance the robustness of the study's conclusions.
2. The study's alignment with current practices in LLMs usage, without significant advancements or novel contributions, may limit its impact on future research.

---

> ### Author Rebuttal · Authors · 2024-05-28
>
> We thank the reviewer for the positive feedback on our work. Below we address some of R4’s concerns:
>
> 1. While OntoNotes has been a valuable resource for coreference research, its age may limit the generalizability of the findings. Consideration of experiments on newer datasets with more complex mentions and coreference examples would enhance the robustness of the study's conclusions.
>
> -> In addition to OntoNotes, we experimented with other coreference datasets in Section 4, which demonstrated the generalizability of findings to a diverse range of domains, languages, and time periods. Moreover, if accepted, we will experiment with more recent coreference datasets, such as CODI-CRAC 2022.

---

> > ### Comment · Reviewer_3vsC · 2024-06-05
> >
> > If extra experiments are carried out I'd be happy for the paper to be accepted

---

### Official Review · Reviewer_P4nJ · 2024-05-09

**Rating:** 5
**Confidence:** 4
**Ethics Flag:** 1

**Summary:**

The paper presents coref experiments using prompted LLMs. Two prompts are considered, one QA-based and the other framed as clustering all given mentions in the document. The latter is found to be clearly more performant. LLMs are given either gold or predicted mentions (using an off-the-shelf manual coref system dcoref); they're not competitive at mention finding themselves, which is likely due to the specific nature of mentions in OntoNotes and other coref datasets. In-context examples are not used (i.e., it's zero-shot) because they're not helpful for the document-level prompt.

The main results are: (1) when dcoref-predicted mentions are used (document-level prompt, zero-shot), strong LLMs like InstructGPT and GPT-4 beat an unsupervised approach weak-SpanBERT which is a SpanBERT trained on the output of dcoref, but fall behind supervised systems, (2) when gold mentions are used, they perform better than supervised and other baselines that also use gold mentions. The paper also performs various transfer and multilingual coref experiments, and find that LLMs generalize well.

**Reasons To Accept:**

- As the paper states, while coref has been used to demonstrate the language understanding capabilities of LLMs since the beginning, there has been no work that reports full coref results (from prompting LLMs), so the results here are undoubtedly valuable as a reference point.
- The paper identifies an effective prompting strategy (document-level, zero-shot), which may not be obvious.
- Experiments are comprehensive, covering the main dataset (OntoNotes) and transfer/multilingual datasets.

**Reasons To Reject:**

- The prompting results are, as acknowledged in the paper and not surprisingly, not amazing compared to SOTA results. This furthermore has various downsides, such as a dependence on a third-party mention finder. Other than for checking off a box on the list of things to verify for LLMs, it's a little trick to extract a lot of lasting technical value from this work.
- The finding that gold mentions allow for amazing coref peformance is known. A supervised system can get ~95 F1 if given gold mentions, this result can be found in [1] but I think the general fact that the coref performance bottleneck is mention finding is a bit of an open secret.
- It seems weak-SpanBERT is the only unsupervised (or distantly supervised) baseline. I don't have strong suggestions for specific additional baselines, but having dcoref and dcoref-based SpanBERT as the only non-LLM non-supervised baselines seems a bit insufficient to fully contextualize the performance of LLM prompting for coref.

[1] Seq2seq is All You Need for Coreference Resolution (Zhang et al., 2023)

---

> ### Author Rebuttal · Authors · 2024-05-28
>
> We thank the reviewer for the feedback on our work. Below we address some of R3’s concerns:
>
> 1. The prompting results are, as acknowledged in the paper and not surprisingly, not amazing compared to SOTA results. This furthermore has various downsides, such as a dependence on a third-party mention finder.
>
> -> While it is true that prompting LLMs for coreference does not outperform SOTA supervised results, we show that this approach empirically outperforms unsupervised coreference. We believe that this was not a given and trivial research question: Yang et al. (2022) shows that coreference resolution was a challenging task for GPT-2 when prompted with multiple-choice templates, and our initial investigations demonstrate that frequently-used QA prompts did not surpass unsupervised coref baseline, even with in-context demonstrations on larger models (Table 9, Appendix A1). In addition, we demonstrated that our Document-level prompting technique is also a robust strategy for coreference resolution across domains, languages, and time periods, if no target language or in-domain data is available for fine-tuning.
>
> Yang et al. 2022. What GPT knows about who is who.  In Proceedings of the Third Workshop on 739 Insights from Negative Results in NLP.
>
> 2. The finding that gold mentions allow for amazing coref peformance is known. A supervised system can get ~95 F1 if given gold mentions, this result can be found in [1] but I think the general fact that the coref performance bottleneck is mention finding is a bit of an open secret.
>
> -> While it may be true that “coref performance bottleneck is mention finding is a bit of an open secret,” previous work only verified this in supervised setting. In contrast, we verified that this finding also holds when prompting LLMs for coreference. We believe that this insight is relevant for future research in prompting LLMs for this task.
>
> 3. It seems weak-SpanBERT is the only unsupervised (or distantly supervised) baseline. I don't have strong suggestions for specific additional baselines, but having dcoref and dcoref-based SpanBERT as the only non-LLM non-supervised baselines seems a bit insufficient to fully contextualize the performance of LLM prompting for coref.
>
> -> As far as we are aware, dcoref and weak-spanBERT are the only existing non-LLM, non-supervised baselines in the coreference literature. We would be happy to compare to other existing baselines in the camera-ready version, if existed.

---

### Official Review · Reviewer_WtZQ · 2024-05-14

**Rating:** 7
**Confidence:** 5
**Ethics Flag:** 1

**Summary:**

This paper reports on experiments evaluating prompt-based approaches to coreference. The approach is very systematic, comparing a number of open-source and proprietary LLMs with unsupervised and supervised models. The main evaluation in Section 3 is carried out on the standard CONLL2012 dataset, but in Section 4 other English datasets are considered, as well as datasets in other languages including  Arabic and Chinese Ontonotes and the SEMEVAL 2010 datasets. The main results for English are that the performance of LLMs on predicted mentions is much lower than that of state-of-the-art coref systems, but on gold mentions is comparable or better. Fine-tuning the LLMs improves the results. On other datasets, InstructGPT performs better out-of-the-box than supervised models trained on other domains, but not than supervised models fine-tuned on a specific domain. The same happens with other languages.

The main contributions / results are:
1. Two prompting strategies are proposed and tested, Question-Answering based and 'Document Template' based (aka document annotation)
2. The very systematic evaluation will be useful to others.

**Questions To Authors:**

Questions:

1. Why didn't you use some of the datasets that appeared more recently? The only recent-ish dataset is LitBank which came out in 2020 - e.g., what about the CODI-CRAC 2022 dataset, which would have also covered more dialogue? Or GUM v. 10?

Suggestions:

1. If accepted, the authors should include a comparison of their results with those of Gan et al 2024, which in my view is the most related work

**Reasons To Accept:**

1. This paper reports on a very systematic evaluation across domains and across languages, presenting many useful results about large-scale evaluation of LLMs for coreference. It is nicely complementary to the paper by Gan et al to be presented at COLING next week, which focuses more on datasets other than CONLL2012 and on hand-evaluation of the LLMs' results, and confirming their results on the effect of mention detection on the performance of LLMs:

Yujian Gan, Massimo Poesio and Juntao Yu (2024). Assessing the Capabilities of Large Language Models in Coreference: An Evaluation. Proc. of LREC/COLING

**Reasons To Reject:**

1. The main weakness of this paper is that it is heavily experimental, not really introducing new ideas but focusing primarily on the evaluation side of things. (Which doesn't make it less useful.)

2. My main concern is the extent to which the results in Section 3 especially are affected by data contamination, given that Ontonotes-2012, Semeval-2010, etc have been around for over than years. But the results e.g., in Section 4.3 make it sound like it can't all just be data contamination.

---

> ### Author Rebuttal · Authors · 2024-05-28
>
> We thank the reviewer for the positive feedback on our work. Below we address some of R2’s questions:
>
> 1. My main concern is the extent to which the results in Section 3 especially are affected by data contamination, given that Ontonotes-2012, Semeval-2010, etc have been around for over than years.
>
> -> As we acknowledged in the limitation section, we do not know with certainty that the evaluated datasets (e.g. OntoNotes) were included in GPT-4’s training data. However, as pointed out by the reviewer, we did provide strong evidence against this case in Section 4.3, where we evaluated the LLMs on a dataset (WSJ-2023) collected after the training date cutoff of the models.
>
> 2. Why didn't you use some of the datasets that appeared more recently? The only recent-ish dataset is LitBank which came out in 2020 - e.g., what about the CODI-CRAC 2022 dataset, which would have also covered more dialogue? Or GUM v. 10? If accepted, the authors should include a comparison of their results with those of Gan et al 2024, which in my view is the most related work
>
> -> The most recent dataset in our work is WSJ-2023, as mentioned above. However, we agreed that it would be helpful to compare with more recent datasets, such as CODI-CRAC 2022 or GUM v. 10. If accepted, we will include the results of these datasets and compare them to Gan et al. 2024, as suggested.

---

> > ### Comment · Reviewer_WtZQ · 2024-06-04
> >
> > Good rebuttal, if those modifications are carried out I'd be happy for the paper to be accepted

---

### Official Review · Reviewer_pW8U · 2024-05-22

**Rating:** 8
**Confidence:** 4
**Ethics Flag:** 1

**Summary:**

This paper evaluates LLMs in their ability to perform coreference resolution using a traditional NLP framework of evaluation and dataset. For coreference resolution, the authors argues that prior work has mostly focused on few-shot learning for sentence-level, syntactically simple coreference datasets such as Winograd Schema Challenge and for pronoun resolution on clinical data. This paper therefore bridges a gap between the LLM literature and coreference resolution literature.

Quality: this paper is overall high quality. The research question is well motivated and bridges a gap between LLMs and traditional NLP based coreference systems. The experiments are well designed with appropriate data benchmarks. The selection of additional research questions is also comprehensive and relevant.

Clarity: this paper is very clear.

Originality: this paper provides a grounded and principled framework to compare prompt-based LLMs and traditional task based (deep learning based and rule based) linguistic models on a well established NLP task. It bridges a gap that shows it's originality.

Significance: medium-high please see above

**Questions To Authors:**

n/a

**Reasons To Accept:**

I especially appreciate the authors doing a good job to provide a grounded and principled framework to compare prompt-based LLMs and traditional task based (deep learning based and rule based) linguistic models on a well established NLP task. It shows a paradigm where we can approach and evaluate LLMs in ways that relate to the traditional task oriented models. It also provides recommendations to the practitioners about when to choose LLMs vs. training a model.

**Reasons To Reject:**

n/a

---

> ### Author Rebuttal · Authors · 2024-05-28
>
> We thank the reviewer for the positive feedback on our work, particularly on the quality, originality, and clarity of our paper. We also appreciate the reviewer’s recognition of the paper’s contributions in providing a principled framework for evaluating prompt-based LLMs on the task of coreference resolution. We hope that our work can further advance LLM research for the task.

---

### Decision · Program_Chairs · 2024-07-10

**Decision:**

Accept

**Comment:**

The paper studies LLMs' ability to perform classic coreference resolution as defined by OntoNotes. The empirical process is solid, and the evidence supports the claims. Having said that, the research question seems less interesting in 2024; as reviewers P4nJ and 3vsC note, it is probably not going to significantly impact future research in language modeling.

[comments from the PCs] We encourage the authors to consider the framing/discussion/conclusion of this paper to show its pertinance to contemporary research and future challenges. This can enhance the paper's reception.